# Enhancing the Accuracy and Fairness of Human Decision Making

**Isabel Valera**[*]
MPI for Intelligent Systems
ivalera@tue.mpg.de

**Adish Singla**[†]
MPI-SWS
adishs@mpi-sws.org

**Manuel Gomez-Rodriguez**[‡]
MPI-SWS
manuelgr@mpi-sws.org

## Abstract

Societies often rely on human experts to take a wide variety of decisions affecting their members, from *jail-or-release* decisions taken by judges and *stop-and-frisk* decisions taken by police officers to *accept-or-reject* decisions taken by academics. In this context, each decision is taken by an expert who is typically *chosen uniformly at random* from a pool of experts. However, these decisions may be imperfect due to limited experience, implicit biases, or faulty probabilistic reasoning. Can we improve the accuracy and fairness of the overall decision making process by optimizing the assignment between experts and decisions?

In this paper, we address the above problem from the perspective of sequential decision making and show that, for different fairness notions in the literature, it reduces to a sequence of (constrained) weighted bipartite matchings, which can be solved efficiently using algorithms with approximation guarantees. Moreover, these algorithms also benefit from posterior sampling to actively trade off exploitation—selecting expert assignments which lead to accurate and fair decisions—and exploration—selecting expert assignments to learn about the experts' preferences. We demonstrate the effectiveness of our algorithms on both synthetic and real-world data and show that they can significantly improve both the accuracy and fairness of the decisions taken by pools of experts.

## 1 Introduction

In recent years, there have been increasing concerns about the potential for unfairness of algorithmic decision making. Moreover, these concerns have been often supported by empirical studies, which have provided, *e.g.*, evidence of racial discrimination [8, 10]. As a consequence, there have been a flurry of work on developing computational mechanisms to make sure that the machine learning methods that fuel algorithmic decision making are fair [3, 4, 5, 6, 13, 14, 15]. In contrast, to the best of our knowledge, there is a lack of machine learning methods to ensure accuracy and fairness in human decision making, which is still prevalent in a wide range of critical applications such as, *e.g.*, jail-or-release decisions by judges, stop-and-frisk decisions by police officers or accept-or-reject decisions by academics. In this work, we take a first step towards filling this gap.

More specifically, we focus on a problem setting that fits a variety of real-world applications, including the ones mentioned above: binary decisions come sequentially over time and each decision need to be taken by a human decision maker, typically an *expert*, who is chosen from a pool of experts. For example, in jail-or-release decisions, the expert is a judge who needs to decide whether she grants bail to a defendant; in stop-and-frisk decisions, the expert is a police officer who needs to decide whether she stop (and potentially frisk) a pedestrian; or, in accept-or-reject decisions, the expert is an academic who needs to decide whether a paper is accepted in a conference (or a journal). In this context, our goal is then to find the optimal assignments between human decision makers and

---

[*]Max Planck Institute for Intelligent Systems. Max Planck Ring 4, 472076 Tuebingen (Germany).

[†]Max Planck Institute for Software Systems (MPI-SWS). Campus E1 5, 66123 Saarbruecken (Germany).

[‡]Max Planck Institute for Software Systems. Paul-Ehrlich-Strasse, G26, 67663 Kaiserslautern (Germany).

decisions which maximizes the accuracy of the overall decision making process while satisfying several popular notions of fairness studied in the literature.

In this paper, we represent human decision making using threshold decisions rules [3] and then show that, if the thresholds used by each expert are known, the above problem can be reduced to a sequence of matching problems, which can be solved efficiently with approximation guarantees. More specifically:

I. Under no fairness constraints, the problem can be cast as a sequence of maximum weighted bipartite matching problems, which can be solved exactly in polynomial (quadratic) time [12].

II. Under (some of the most popular) fairness constraints, the problem can be cast as a sequence of bounded color matching problems, which can be solved using a bi-criteria algorithm based on linear programming techniques with a $1/2$ approximation guarantee [9].

Moreover, if the thresholds used by each expert are unknown, we also show that, if we estimate the value of each threshold using posterior sampling, we can effectively trade off exploitation—taking accurate and fair decisions—and exploration—learning about the experts' preferences and biases. More formally, we can show that posterior samples achieve a sublinear regret in contrast to point estimates, which suffer from linear regret.

Finally, we experiment on synthetic data and real jail-or-release decisions by judges [8]. The results show that: (i) our algorithms improve the accuracy and fairness of the overall human decision making process with respect to random assignment; (ii) our algorithms are able to ensure fairness more effectively if the pool of experts is diverse, *e.g.*, there exist harsh judges, lenient judges, and judges in between; and, (iii) our algorithms are able to ensure fairness even if a significant percentage of judges (*e.g.*, 50%) has preferences (biases) against a group of individuals sharing a certain sensitive attribute value (*e.g.*, race). The implementations of our algorithms and the data used in our experiments are available at `https://github.com/Networks-Learning/FairHumanDecisions`.

## 2 Preliminaries

In this section, we first define decision rules and formally define their utility and group benefit. Then, we revisit threshold decision rules, a type of decision rules which are optimal in terms of accuracy under several notions of fairness from the literature.

**Decision rules, their utilities, and their group benefits.** Given an individual with a feature vector $\boldsymbol{x} \in \mathbb{R}^d$, a (*ground-truth*) label $y \in \{0, 1\}$, and a sensitive attribute $z \in \{0, 1\}$, a decision rule $d(\boldsymbol{x}, z) \in \{0, 1\}$ controls whether the ground-truth label $y$ is *realized* by means of a binary decision about the individual. As an example, in a pretrial release scenario, the decision rule specifies whether the individual remains in jail, *i.e.*, $d(\boldsymbol{x}, z) = 1$ if she remains in jail and $d(\boldsymbol{x}, z) = 0$ otherwise; the label indicates whether a released individual would reoffend, *i.e.*, $y = 1$ if she would reoffend and $y = 0$ otherwise; the feature vector $\boldsymbol{x}$ may include the current offense, previous offenses, or times she failed to appear in court; and the sensitive attribute $z$ may be race, *i.e.*, black vs white.

Further, we define random variables $X$, $Y$, and $Z$ that take on values $X = x$, $Y = y$, and $Z = z$ for an individual drawn randomly from the population of interest. Then, we measure the (immediate) utility as the overall profit obtained by the decision maker using the decision rule [3], *i.e.*,

$$u(d, c) = \mathbb{E}\left[Y d(X, Z) - c\, d(X, Z)\right] = \mathbb{E}\left[d(X, Z)\left(P_{Y|X,Z} - c\right)\right] \tag{1}$$

where $c \in (0, 1)$ is a given constant. For example, in a pretrial release scenario, the first term is proportional to the expected number of violent crimes prevented under $d$, the second term is proportional to the expected number of people detained, and $c$ measures the cost of detention in units of crime prevented. Here, note that the above utility reflects only the proximate costs and benefits of decisions rather than long-term, systematic effects. Finally, we define the (immediate) group benefit as the fraction of beneficial decisions received by a group of individuals sharing the sensitive attribute value $z$ [15], *i.e.*,

$$b_z(d, c) = \mathbb{E}\left[f(d(X, Z = z))\right]. \tag{2}$$

For example, in a pretrial release scenario, one may define $f(x) = 1 - x$ and thus the benefit to the group of white individuals be proportional to the expected number of them who are released under $d$. Remarkably, most of the notions of (un)fairness used in the literature, such as disparate impact [1], equality of opportunity [6] or disparate mistreatment [13] can be expressed in terms of group benefits. Finally, note that, in some applications, the beneficial outcome may correspond to $d(X, Z) = 1$.

**Optimal threshold decision rules.** Assume the conditional distribution $P(Y|X, Z)$ is given[4]. Then, the optimal decision rules $d^*$ that maximize $u(d, c)$ under the most popular fairness constraints from the literature are threshold decision rules [3, 6]:

— *No fairness constraints*: the optimal decision rule under no fairness constraints is given by the following deterministic threshold rule:

$$d^*(X, Z) = \begin{cases} 1 & \text{if } p_{Y=1|X,Z} \geq c \\ 0 & \text{otherwise.} \end{cases} \tag{3}$$

— *Disparate impact, equality of opportunity, and disparate mistreatment*: the optimal decision rule which satisfies (avoids) the three most common notions of (un)fairness is given by the following deterministic threshold decision rule:

$$d^*(X, Z) = \begin{cases} 1 & \text{if } p_{Y=1|X,Z} \geq \theta_Z \\ 0 & \text{otherwise,} \end{cases} \tag{4}$$

*where $\theta_Z \in [0, 1]$ are constants that depend only on the sensitive attribute and the fairness notion of interest. Note that the unconstraint optimum can be also expressed using the above form if we take $\theta_Z = c$.*

## 3 Problem Formulation

In this section, we first use threshold decision rules to represent biased human decisions and then formally define our sequential human decision making process.

**Humans as threshold decision rules.** Inspired by recent work by Kleinberg et al. [7], we model a human decision maker $v$ who has access to $p_{Y|X,Z}$ using the following threshold decision rule:

$$d_v(X, Z) = \begin{cases} 1 & \text{if } p_{Y=1|X,Z} \geq \theta_{V,Z} \\ 0 & \text{otherwise,} \end{cases} \tag{5}$$

where $\theta_{V,Z} \in [0, 1]$ are constants that depend on the decision maker and the sensitive attribute, and they represent human decision makers' *biases* (or *preferences*) towards groups of people sharing a certain value of the sensitive attribute $z$. For example, in a pretrial release scenario, if a judge $v$ is generally more lenient towards white people ($z = 0$) than towards black people ($z = 1$), then $\theta_{v,z=0} > \theta_{v,z=1}$.

In the above formulation, note that we assume all experts make predictions using the same (true) conditional distribution $p_{Y|X,Z}$, *i.e.*, all experts have the same prediction ability. It would be very interesting to relax this assumption and account for experts with different prediction abilities. However, this entails a number of non trivial challenges and is left for future work.

**Sequential human decision making problem.** A set of human decision makers $\mathcal{V} = \{v_k\}_{k \in [n]}$ need to take decisions about individuals over time. More specifically, at each time $t \in \{1, \ldots, T\}$, there are $m$ decisions to be taken and each decision $i \in [m]$ is taken by a human decision maker $v_i(t) \in \mathcal{V}$, who applies her threshold decision rule $d_{v_i(t)}(X, Z)$, defined by Eq. 5, to the corresponding feature vector $x_i(t)$ and sensitive attribute $z_i(t)$. Note that we assume that $y_i(t)|x_i(t), z_i(t) \sim p_{Y|X,Z}$ for all $t \in \{1, \ldots, T\}$ and $i \in [m]$.

At each time $t$, our goal is then to find the assignment of human decision makers $v_i(t)$ to individuals $(x_i(t), z_i(t))$, with $v_i(t) \neq v_j(t)$ for all $i \neq j$, that maximizes the expected utility of a sequence of decisions, *i.e.*,

$$u_{\leq T}(\{d_{v_i(t)}\}_{i,t}, c) = \frac{1}{mT} \sum_{t=1}^{T} \sum_{i=1}^{m} d_{v_i(t)}(x_i(t), z_i(t))(p_{Y=1|x_i(t),z_i(t)} - c), \tag{6}$$

where $u_{\leq T}(\{d_{v_i(t)}\}_{i,t}, c)$ is a empirical estimate of a straight forward generalization of the utility defined by Eq. 1 to multiple decision rules.

## 4 Proposed Algorithms

In this section, we formally address the problem defined in the previous section without and with fairness constraints. Note that, without fairness constraints, we aim to approximate the solution provided by Eq. 3, and with fairness constraints, Eq. 4. In both cases, we first consider the setting in which the human decision makers' thresholds $\theta_{V,Z}$ are known and then generalize our algorithms to the setting in which they are unknown and need to be learned over time.

**Decisions under no fairness constraints.** We can find the assignment of human decision makers $\{v(t)\}_{t=1}^{T}$ with the highest expected utility by solving the following optimization problem:

$$\text{maximize} \quad \sum_{t=1}^{T} \sum_{i=1}^{m} d_{v_i(t)}(x_i(t), z_i(t))(p_{Y=1|x_i(t),z_i(t)} - c)$$
$$\text{subject to} \quad v_i(t) \in \mathcal{V} \text{ for all } t \in \{1, \dots, T\},$$
$$v_i(t) \neq v_j(t) \text{ for all } i \neq j. \tag{7}$$

— *Known thresholds.* If the thresholds $\theta_{V,Z}$ are known for all human decision makers, the above problem decouples into $T$ independent subproblems, one per time $t \in \{1, \dots, T\}$, and each of these subproblems can be cast as a maximum weighted bipartite matching, which can be solved exactly in polynomial (quadratic) time [12]. To do so, for each time $t$, we build a weighted bipartite graph where each human decision maker $v_j$ is connected to each individual $(x_i(t), z_i(t))$ with weigh:

$$w_{ji} = \begin{cases} p_{Y|x_i(t),z_i(t)} - c & \text{if } p_{Y|x_i(t),z_i(t)} \geq \theta_{v_j,z_i(t)} \\ 0 & \text{otherwise,} \end{cases}$$

Note that the maximum weighted bipartite matching is the optimal assignment as defined by Eq. 7.

— *Unknown thresholds.* If the thresholds are unknown, we need to trade off exploration, *i.e.*, learning about the thresholds $\theta_{V,Z}$, and exploitation, *i.e.*, maximizing the average utility. To this aim, for every decision maker $v$, we assume a Beta prior over each threshold $\theta_{v,z} \sim Beta(\alpha, \beta)$. Under this assumption, after round $t$, we can update the (domain of the) distribution of $\theta_{v,z}(t)$ as:

$$\max(0, \theta_{v,z}^{L}(t)) \leq \theta_{v,z}(t) \leq \min(1, \theta_{v,z}^{H}(t)), \tag{8}$$

where

$$\theta_{v,z}^{L}(t) = \max_{t' \leq t \,|\, z_i(t')=z,\, v_i(t')=v,\, d_v(x_i(t'),z)=0} p_{Y=1|x_i(t'),z}$$
$$\theta_{v,z}^{H}(t) = \min_{t' \leq t \,|\, z_i(t')=z,\, v_i(t')=v,\, d_v(x_i(t'),z)=1} p_{Y=1|x_i(t'),z},$$

and write the posterior distribution of $\theta_{v,z}(t)$ as

$$p(\theta_{v,z}(t)|\mathcal{D}(t)) = \frac{\Gamma(\alpha+\beta)(\theta_{v,z}^{H}(t) - \theta_{v,z}(t))^{\alpha-1}(\theta_{v,z}(t) - \theta_{v,z}^{L}(t))^{\beta-1}}{\Gamma(\alpha)\Gamma(\beta)(\theta_{v,z}^{H}(t) - \theta_{v,z}^{L}(t))^{\alpha+\beta-1}}, \tag{9}$$

Then, at the beginning of round $t+1$, one can think of estimating the value of each threshold $\theta_{v,z}(t)$ using point estimates, *i.e.*, $\hat{\theta}_{v,z} = \text{argmax} \, p(\theta_{v,z}(t)|\mathcal{D}(t))$, and use the same algorithm as for known thresholds. Unfortunately, if we define regret as follows:

$$R(T) = u_{\leq T}(\{d_{v_i(t)}\}_{i,t}, c) - u_{\leq T}(\{d_{v_i^*(t)}\}_{i,t}, c), \tag{10}$$

where $v_i(t)$ is the optimal assignment under the point estimates of the thresholds and $v_i^*(t)$ is the optimal assignment under the true thresholds, we can show that (proven in Appendix A):

**Proposition 1** *The optimal assignments with deterministic point estimates for the thresholds suffers linear regret $\Theta(T)$.*

The above result is a consequence of insufficient exploration, which we can overcome if we estimate the value of each threshold $\theta_{v,z}(t)$ using posterior sampling, *i.e.*, $\hat{\theta}_{v,z} \sim p(\theta_{v,z}(t)|\mathcal{D}(t))$, as formalized by the following theorem:

**Theorem 2** *The expected regret of the optimal assignments with posterior samples for the thresholds is $O(\sqrt{T})$.*

*Proof Sketch.* The proof of this theorem follows via interpreting the problem setting as a reinforcement learning problem. Then, we can apply the generic results for reinforcement learning via posterior sampling of [11]. In particular, we map our problem to an MDP with horizon 1 as follows. The actions in the MDP correspond to assigning $m$ individuals to $n$ experts (given by $K$) and the reward is given by the utility at time $t$.

Then, it is easy to conclude that the expected regret of the optimal assignments with posterior samples for the thresholds is $O(S\sqrt{KT\log(SKT)})$, where $K = n.(n-1).(n-2)\ldots(n-m+1)$ denotes the possible assignments of $m$ individuals to $n$ experts and $S$ is a problem dependent parameter. $S$ quantifies the the total number of states/realizations of feature vectors $x_i$ and sensitive features $z_i$ to the $i \in [m]$ individuals—note that $S$ is bounded only for the setting where feature vectors $x_i$ and sensitive features $z_i$ are discrete.

Given that the regret only grows as $O(\sqrt{T})$ (i.e., sublinear in $T$), this theorem implies that the algorithm based on optimal assignments with posterior samples converges to the optimal assignments given the true thresholds as $T \to \infty$.

**Decisions under fairness constraints.** For ease of exposition, we focus on disparate impact, however, a similar reasoning follows for equality of opportunity and disparate mistreatment [6, 13].

To avoid disparate impact, the optimal decision rule $d^*(X, Z)$, given by Eq. 4, maximizes the utility, as defined by Eq. 1, under the fairness constraint [3, 13] $DI(d^*, c) = |b_{z=1}(d^*, c) - b_{z=0}(d^*, c)| \leq \alpha$, where $\alpha \in [0, 1]$ is a given parameter which controls the *amount* of disparate impact—the smaller the value of $\alpha$, the lower the disparate impact of the corresponding decision rule. Similarly, we can calculate a empirical estimate of the disparate impact of a decision rule $d$ at each time $t$ as:

$$DI_t(d, c) = \frac{1}{m}|b_{t,z=1}(d, c) - b_{t,z=0}(d, c)|, \tag{11}$$

where $b_{t,z}(d, c) = \sum_{i=1}^{m} \mathbb{I}(z_i = z)f(d(x_i(t), z_i(t)))$, where $f(\cdot)$ defines what is a beneficial outcome. Here, it is easy to see that, for the optimal decision rule $d^*$ under impact parity, $DI_t(d^*, c)$ converges to $DI(d^*, c)$ as $m \to \infty$, and $1/T\sum_{t=1}^{T} DI_t(d^*, c)$ converges to $DI(d^*, c)$ as $T \to \infty$.

For a fixed $\alpha$, assume there are at least $m(1 - \alpha)$ experts with $\theta_{v,z} < c$, at least $m(1 - \alpha)$ experts with $\theta_{v,z} \geq c$ for each $z = 0, 1$, and $n \geq 2m$. Then, we can find the assignment of human decision makers $\{v(t)\}$ with the highest expected utility and disparate impact less than $\alpha$ as:

$$\text{maximize} \quad \sum_{t=1}^{T}\sum_{i=1}^{m} d_{v_i(t)}(x_i(t), z_i(t))(p_{Y=1|x_i(t),z_i(t)} - c),$$
$$\text{subject to} \quad v_i(t) \in \mathcal{V} \text{ for all } t \in \{1, \ldots, T\},$$
$$v_i(t) \neq v_j(t) \text{ for all } i \neq j,$$
$$b_{t,z}(d^*, c) - \alpha m_z(t) \leq b_{t,z}(\{d_{v_i(t)}\}_i) \,\forall t, z$$
$$b_{t,z}(\{d_{v_i(t)}\}_i) \leq b_{t,z}(d^*, c) + \alpha m_z(t) \,\forall t, z. \tag{12}$$

where and $m_z(t)$ is the number of decisions with sensitive attribute $z$ at round $t$ and $b_{t,z}(\{d_{v_i(t)}\}_i) = \sum_{i=1}^{m} \mathbb{I}(z_i = z)f(d_{v_i(t)}(x_i(t), z_i(t)))$. Here, the assignment $v^*(t)$ given by the solution to the above optimization problem satisfies that $DI_t(\{d_{v_i^*(t)}\}_i, c) \in [DI_t(d^*, c) - \alpha, DI_t(d^*, c) + \alpha]$ and thus $\lim_{T \to \infty} DI_{\leq T}(\{d_{v_i^*(t)}\}_{i,t}, c) \leq \alpha$.

— *Known thresholds.* If the thresholds are known, the problem decouples into $T$ independent subproblems, one per time $t \in \{1, \ldots, T\}$, and each of these subproblems can be cast as a constrained maximum weighted bipartite matching. To do so, for each time $t$, we build a weighted bipartite graph where each human decision maker $v_j$ is connected to each individual $(x_i(t), z_i(t))$ with weight $w_{ji}$, where

$$w_{ji} = \begin{cases} p_{Y|x_i(t),z_i(t)} - c & \text{if } p_{Y|x_i(t),z_i(t)} \geq \theta_{v_j,z_i(t)} \\ 0 & \text{otherwise,} \end{cases}$$

and we additionally need to ensure that, for $z \in \{0, 1\}$, the matching $\mathcal{S}$ satisfies that

$$b_{t,z}(d^*, c) - \alpha m_z(t) \leq \sum_{(j,i)\in\mathcal{S}:z_i=z} g(w_{ji}) \quad \text{and} \quad \sum_{(j,i)\in\mathcal{S}:z_i=z} g(w_{ji}) \leq b_{t,z}(d^*, c) + \alpha m_z(t),$$

where $m_z(t)$ denotes the number of individuals with sensitive attribute $z$ at round $t$ and the function $g(w_{ji})$ depends on what is the beneficial outcome, *e.g.*, in a pretrial release scenario, $g(w_{ji}) = \mathbb{I}(w_{ji} \neq 0)$. Remarkably, we can reduce the above constrained maximum weighted bipartite matching problem to an instance of the bounded color matching problem [9], which allows for a bi-criteria algorithm based on linear programming techniques with a $1/2$ approximation guarantee. To do so, we just need to rewrite the above constraints as

$$\sum_{(j,i)\in\mathcal{S}:z_i=z} g(w_{ji}) \leq b_{t,z}(d^*,c) + \alpha m_z(t), \quad \text{and} \tag{13}$$

$$\sum_{(j,i)\in\mathcal{S}:z_i=z} g^C(w_{ji}) \leq (1+\alpha)m_z(t) - b_{t,z}(d^*,c). \tag{14}$$

To see the equivalence between the above constraints and the original ones, one needs to realize that we are looking for a perfect matching and thus $\sum_{(j,i)\in\mathcal{S}:z_i=z}\left[g(w_{ji}) + g^C(w_{ji})\right] = m_z(t)$. For example, in a pretrial release scenario, $g(w_{ji}) = \mathbb{I}(w_{ji} \neq 0)$ and $g^C(w_{ji}) = \mathbb{I}(w_{ji} = 0)$.

— *Unknown thresholds.* If the threshold are unknown, we proceed similarly as in the case under no fairness constraints, *i.e.*, we again assume Beta priors over each threshold, update their posterior distributions after each time $t$, and use posterior sampling to set their values at each time.

Finally, for the regret analysis, we focus on an alternative unconstrained problem, which is equivalent to the one defined by Eq. 12 by Lagrangian duality [2]:

$$\begin{aligned}
\text{maximize} \quad & \sum_{t=1}^{T}\sum_{i=1}^{m} d_{v_i(t)}(x_i(t), z_i(t))(p_{Y=1|x_i(t),z_i(t)} - c) \\
& + \sum_{t=1}^{T}\sum_{i=1}^{m} \lambda_{l,t,z}\left(b_{t,z}(d^*,c) - b_{t,z}(\{d_{v_i(t)}\}_i) - \alpha m_z(t)\right) \\
& + \sum_{t=1}^{T}\sum_{i=1}^{m} \lambda_{u,t,z}\left(b_{t,z}(\{d_{v_i(t)}\}_i) - b_{t,z}(d^*,c) - \alpha m_z(t)\right) \\
\text{subject to} \quad & v_i(t) \in \mathcal{V} \text{ for all } t \in \{1,\dots,T\}, \\
& v_i(t) \neq v_j(t) \text{ for all } i \neq j.
\end{aligned} \tag{15}$$

where $\lambda_{l,t,z} \geq 0$ and $\lambda_{u,t,z} \geq 0$ are the Lagrange multipliers for the band constraints. Then, we can then state the following theoretical result (the proof easily follows from the proof of Theorem 2):

**Theorem 3** *The expected regret of the optimal assignments for the problem defined by Eq. 15 with posterior samples for the thresholds is $O(\sqrt{T})$.*

**Remark.** In the above formulation, we do not enforce a specific mechanism to reduce disparate impact–our framework finds the solution with maximum utility that satisfies the disparate impact constraint. Depending on the distribution $p_{Y|X,Z}$ and the definition of utility and benefits, such a solution will result in an increase (decrease) of release rates for group $z = 0$ ($z = 1$) or viceversa.

## 5 Experiments

In this section we empirically evaluate our framework on both synthetic and real data. To this end, we compare the performance, in terms of both utility and fairness, of the following algorithms:

— *Optimal:* Every decision is taken using the optimal decision rule $d^*$, which is defined by Eq. 3 under no fairness constraints and by Eq. 4 under fairness constraints.

— *Known:* Every decision is taken by a judge following a (potentially biased) decision rule $d_v$, as given by Eq. 5. The threshold for each judge is known and the assignment between judges and decisions is found by solving the corresponding matching problem, *i.e.*, Eq, 7 under no fairness constraints and Eq. 12 under fairness constraints.

— *Unknown:* Every decision is taken by a judge following a (potentially biased) decision rule $d_v$, proceeding similarly as in "Known". However, the threshold for each judge is unknown it is necessary to use posterior sampling to estimate the thresholds.

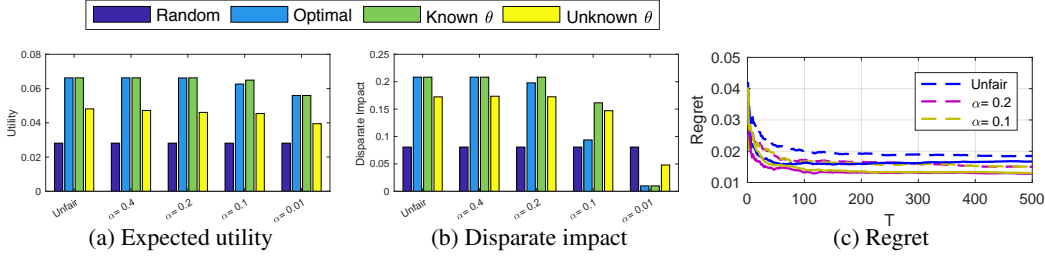

Figure 1: Performance in synthetic data. Panels (a) and (b) show the trade-off between expected utility and disparate impact. For the utility, the higher the better and, for the disparate impact, the lower the better. Panel (c) shows the regret achieved by our algorithm under unknown experts' thresholds as defined in Eq. 10. Here, the solid lines show the results for $m = 20$ and dashed lines for $m = 10$.

— ***Random:*** Every decision is taken by a judge following a (potentially biased) decision rule $d_v$. The assignment between judges and decision is random.

## 5.1 Experiments on Synthetic Data

**Experimental setup.** For every decision, we first sample the sensitive attribute $z_i \in \{0, 1\}$ from $Bernouilli(0.5)$ and then sample $p_{Y=1|x_i,z_i} \sim Beta(3, 5)$ if $z_i = 0$ and from $p_{Y=1|x_i,z_i} \sim Beta(4, 3)$, otherwise. For every expert, we generate her decision thresholds $\theta_{v,0} \sim Beta(0.5, 0.5)$ and $\theta_{v,1} \sim Beta(5, 5)$. Here we assume there are $n = 3m$ experts, to ensure that in each round there are at least $m(1 - \alpha)$ experts with $\theta_{v,z} < c$, at least $m(1 - \alpha)$ experts with $\theta_{v,z} \geq c$ for $z \in \{0, 1\}$. In practice, if there is no feasible assignment for a round and desired level of fairness $\alpha$, one may decide to: i) add experts to the pool to increase diversity; ii) decrease the number of cases per round; or (iii) use a random assignment in that round. Finally, we set $m = 20$, $T = 1000$ and $c = 0.5$, and the beneficial outcome for an individual is $d = 1$, *i.e.*, $f(d) = d$.

**Results.** Figures 1(a)-(b) show the expected utility and the disparate impact after $T$ units of time for the optimal decision rule and for the group of experts under the assignments provided our algorithms and under random assignments. We find that the experts chosen by our algorithm provide decisions with higher utility and lower disparate impact than the experts chosen at random, even if the thresholds are unknown. Moreover, if the threshold are known, the experts chosen by our algorithm closely match the performance of the optimal decision rule both in terms of utility and disparate impact. Finally, we compute the regret as defined by Eq. 10, *i.e.*, the difference between the utilities provided by algorithm with *Known* and *Unknown* thresholds over time. Figure 1(c) summarizes the results, which show that, as time progresses, the regret degreases at a rate $O(\sqrt{T})$.

## 5.2 Experiments on Real Data

**Experimental setup.** We use the COMPAS recidivism prediction dataset compiled by ProPublica [8], which comprises of information about all criminal offenders screened through the COMPAS (Correctional Offender Management Profiling for Alternative Sanctions) tool in Broward County, Florida during 2013-2014. In particular, for each offender, it contains a set of demographic features (gender, race, age), the offender's criminal history (*e.g.*, the reason why the person was arrested, number of prior offenses), and the risk score assigned to the offender by COMPAS. Moreover, ProPublica also collected whether or not these individuals actually recidivated within two years after the screening.

In our experiments, the sensitive attribute $z \in \{0, 1\}$ is the race (white, black), the label $y$ indicates whether the individual recidivated ($y = 1$) or not ($y = 0$), the decision rule $d$ specifies whether an individual is released from jail ($d = 0$) or not ($d = 1$) and, for each sensitive attribute $z$, we approximate $p_{Y|X,Z=z}$ using a logistic regression classifier, which we train on 25% of the data. Then, we use the remaining 75% of the data to evaluate our algorithm as follows. Since we do not have information about the identify of the judges who took each decision in the dataset, we *create* $N = 3m$ (fictitious) judges and sample their thresholds from a $\theta \sim Beta(\tau, \tau)$, where $\tau$ controls the diversity (lenient vs harsh) across judges by means the standard deviation of the distribution since $std(\theta) = \frac{1}{4\tau(2\tau+1)}$. Here, we consider two scenarios: (i) all experts are unbiased towards race and thus $\theta_{v0} = \theta_{v1}$ and (ii) 50% of the experts are unbiased towards race and the other 50% are biased, *i.e.*, $\theta_{v1} = 1.2\theta_{v0}$. Finally, we consider $m = 20$ decisions per round, which results into 197 rounds, where we assign decisions to rounds at random.

**Results.** Figure 2 shows the expected utility, the true utility and the disparate impact after $T$ units of time for the optimal decision rule and for the group of unbiased experts (scenario (i))

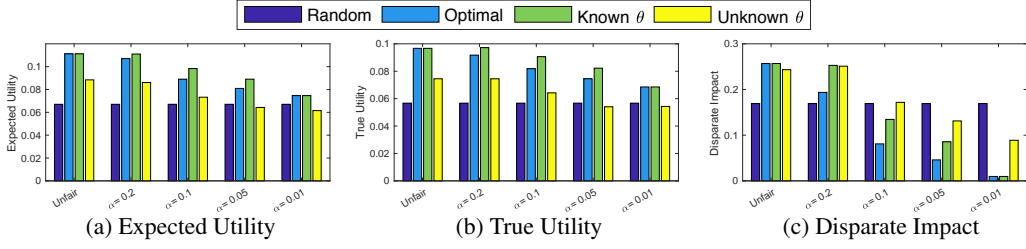

Figure 2: Performance in COMPAS data. Panels (a) and (b) show the expected utility and true utility and panel (c) shows the disparate impact. For the expected and true utility, the higher the better and, for the disparate impact, the lower the better.

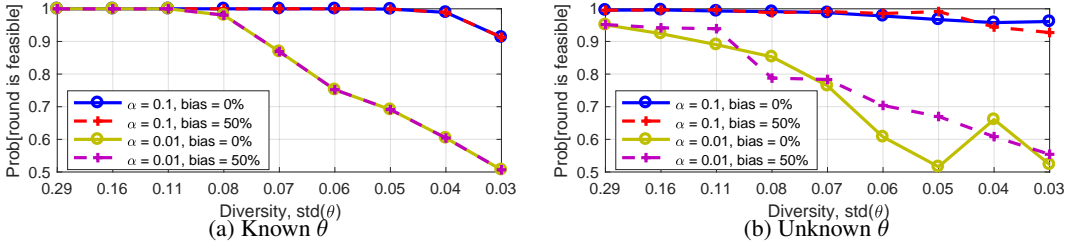

Figure 3: Feasibility in COMPAS data. Probability that a round does not allow for an assignment between judges and decisions with less than $\alpha$ disparate impact for different pools of experts of varying diversity and percentage of biased judges.

under the assignments provided our algorithms and under random assignments. The true utility $\hat{u}_{\leq T}(d, c)$ is just the utility after $T$ units of time given the actual true $y$ values rather than $p_{Y|X,Z}$, *i.e.*, $\hat{u}_{\leq T}(d, c) = \frac{1}{T} \sum_{t=1}^{T} \sum_{i=1}^{m} d(x_i(t), z_i(t))(y_i - c)$. Similarly as in the case of synthetic data, we find that the judges chosen by our algorithm provide higher expected utility and true utility as well as lower disparate impact than the judges chosen at random, even if the thresholds are unknown. We notice that our algorithm achieves a level of fairness $\alpha$ by decreasing the release rate of white defendants, since our definition of utility penalizes more to release individuals who will be more likely to recidivate than to keep in jail those who will not. Note also that under fairness constraints, our algorithm relies on a bi-criteria algorithm with a 1/2 approximation guarantee to solve the maximum weighted bipartite matching problem. As a consequence, it sometimes finds matchings violate the fairness constraints but have higher utility.

Figure 3 shows the probability that a round does not allow for an assignment between judges and decisions with less than $\alpha$ disparate impact for different pools of experts of varying diversity and percentage of biased judges. The results show that, on the one hand, our algorithms are able to ensure fairness more effectively if the pool of experts is diverse and, on the other hand, our algorithms are able to ensure fairness even if a significant percentage of judges (*e.g.*, 50%) are biased against a group of individuals sharing a certain sensitive attribute value.

## 6 Conclusions

In this paper, we have proposed a set of practical algorithms to improve the utility and fairness of a sequential decision making process, where each decision is taken by a human expert, who is selected from a pool experts. Experiments on synthetic data and real jail-or-release decisions by judges show that our algorithms are able to mitigate imperfect human decisions due to limited experience, implicit biases or faulty probabilistic reasoning. Moreover, they also reveal that our algorithms benefit from higher diversity across the pool experts, being able to ensure fairness even if a significant percentage of judges are biased against a group of individuals sharing a sensitive attribute value (*e.g.*, race).

There are many interesting venues for future work. For example, in our work, we assumed all experts make predictions using the same (true) conditional distribution and then apply (potentially) different thresholds. We have also assumed that experts do not learn from the decisions they take over time, *i.e.*, their prediction model and thresholds are fixed. It would be very interesting to relax these assumptions and account for experts with different prediction abilities. In some scenarios, a decision is taking jointly by a group of experts, *e.g.*, faculty recruiting decisions. It would be a natural follow-up to the current work to design our algorithms for such scenario. Finally, in our experiments, we have to generate fictitious judges since we do not have information about the identify of the judges who took each decision. It would be very valuable to gain access to datasets with such information [7].

**Acknowledgments.** Isabel Valera acknowledges funding from a MPG Minerva Fast Track Grant.

## Footnotes

[4]In practice, the conditional distribution may be approximated using a machine learning model trained on historical data.

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

# A  Proof sketch of Proposition 1

Consider a simple setup with $n = 2$ experts and $m = 1$ decision at each round $t \in [T]$. Furthermore, we fix the following two things before setting up the problem instance: (i) let $g(\cdot)$ be a deterministic function which computes a point estimate of a distribution (*e.g.*, mean, or MAP); (ii) we assume a deterministic tie-breaking by the assignment algorithm, and w.l.o.g. expert $j = 1$ is preferred over expert $j = 2$ for assignment when both of them have same edge weights.

For the first expert $j = 1$, we know the exact value of the threshold $\theta_{1,z}$. For the second expert $j = 2$, the threshold $\theta_{2,z}$ could take any value in the range $[0, 1]$ and we are given a prior distribution $p(\theta_{2,z})$. Let us denote $\widetilde{\theta}_{2,z} = g\big(p(\theta_{2,z})\big)$. Now, we construct a problem instance for which the algorithm would suffer linear regret separately for $\widetilde{\theta}_{2,z} > 0$ and $\widetilde{\theta}_{2,z} = 0$.

**Problem instance if $\widetilde{\theta}_{2,z} > 0$**

We consider a problem instance as follows: $c = 0$, $\theta_{2,z} = 0$, $\theta_{1,z} = \frac{c + \widetilde{\theta}_{2,z}}{2}$, and for all $t \in [T]$ we have $p_{Y=1|x_i(t),z_i(t)}$ uniformly sampled from the range $(c, \widetilde{\theta}_{2,z})$ (note that $m = 1$ and there is only one individual $i = 1$ at each round $t$). The algorithm would always assign the individual to expert $j = 1$ and has a cumulative expected utility of $\frac{3T\widetilde{\theta}_{2,z}}{8}$. However, given the true thresholds, the algorithm would have always assigned the individual to expert $j = 2$ and would have a cumulative expected utility of $\frac{T\widetilde{\theta}_{2,z}}{2}$. Hence, the algorithm suffers a linear regret of $R(T) = \frac{T\widetilde{\theta}_{2,z}}{8}$.

**Problem instance if $\widetilde{\theta}_{2,z} = 0$**

We consider a problem instance as follows: $c = 1$, $\theta_{2,z} = 1$, $\theta_{1,z} = \frac{c + \widetilde{\theta}_{2,z}}{2}$, and for all $t \in [T]$ we have $p_{Y=1|x_i(t),z_i(t)}$ uniformly sampled from the range $(\widetilde{\theta}_{2,z}, c)$ (note that $m = 1$ and there is only one individual $i = 1$ at each round $t$). The algorithm would always assign the individual to expert $j = 1$ and has a cumulative expected utility of $\frac{-T\widetilde{\theta}_{2,z}}{8}$. However, given the true thresholds, the algorithm would have always assigned the individual to expert $j = 2$ and would have a cumulative expected utility of $0$. Hence, the algorithm suffers a linear regret of $R(T) = \frac{T\widetilde{\theta}_{2,z}}{8}$.

