[Reviews · NeurIPS 2018]

Reviewer 1



## Updates after feedback/discussion ## The theoretical framework proposed in this paper is interesting and lends itself to numerous extensions that could be pursued in future work. Overall I think the authors convincingly demonstrate that their approach can improve utility over the case of random assignment. The evidence for fairness improvement might run into opposition as the assignment strategy may be viewed as discriminatory in practice. This would be worth further discussion in the paper. It would also be interesting to see more examples of what happens when there aren't all that many more decision makers than items. A typical situation is one where there are more individuals than judges, so each judge needs to be used each round. However, not all individuals are expected to be processed in each round. I also think the authors can do more to provide motivation and intuition for their strategy. Here's my interpretation of the overall approach. The result for the unconstrained optimization setting can be interpreted as approximating the rule in equation (3) through a strategic assignment of judges to individuals. The rule in equation (3) applies the same threshold to everyone regardless of their group. This rule holds everyone, regardless of group, to the same optimal threshold. One can interpret the suboptimality of random assignment as arising from a (judge, individual) combination that differs from the optimal thresholding rules in equations (3). There are two reasons for why the random assignment rule may be suboptimal. (i) If judges are systematically biased against some group, the random assignment rule would effectively apply a higher standard to one group than another. We know that a single common threshold rule is optimal, so random assignment is suboptimal. (ii) Even if judges don't use group-specific thresholds, random assignment may fail to agree with thresholding at the optimal threshold c in equation (3). We can interpret the approach outlined in the paper as bringing us closer to the rule in equation (3) (equal standard) in the unconstrained case. We can interpret the results for the constrained case as bringing us closer (than random assignment) to the rule in equation (4), that optimally holds groups to different standards. ## Original review ## Overview: This paper considers the following problem. Suppose that there are m judges available to make decisions about n < m individuals (per round). Individuals are described by (X, Z, Y), where X are features, Z is a group indicator (binary), and Y is an outcome. Judges make decisions to maximize immediate utility by thresholding the common (and assumed known) conditional probability function p(x, z) = P(Y = 1 | X, Z) at judge-specific and group-specific thresholds. What is the optimal way of assigning individuals to judges in each round so as to maximize utility subject to fairness (DI, TPR, TPR/FPR) constraints? The authors show that the unconstrained problem can be solved efficiently as a bipartite matching between judges and individuals. The constrained problem can be reduced to bounded color matching. Theoretical results are provided showing that posterior sampling can achieve sublinear regret in the cases where the judge-specific thresholds are unknown and need to be estimated across rounds. Empirical evaluations demonstrate performance of the proposed method in comparison to the more common strategy of random assignment of cases to judges. Comments: Overall I think this paper does a pretty good job from a theoretical standpoint of analyzing a toy problem that has some relevance to practice. The authors readily admit that assuming all judges have access to the the same true probability function for decision making is highly idealized. I wish the authors did more in either the discussion of the theory or the empirical results to more clearly characterize where the gains are coming from. In terms of practical implications, the most relevant empirical comparison is between random assignment and the unknown thresholds case. As I elaborate on below, I would like the authors to expand on this comparison to clarify their ideas. - While DI as in (11) depends on there being 2 groups, it seems like for a more general definition of DI the results generalize to to non-binary Z. Is this the case? - The optimal threshold decision rules have the form given in (3) and (4) only if you assume that the probability function p(x,z) has positive density (see original Corbett-Davies paper). If you have point masses in p(x,z), the optimal rule winds up being a randomized one. - Discussion of COMPAS data: Individuals in bail proceedings are considered "defendants", not "offenders". They have not yet been convicted of any crimes, and thus have not formally "offended". - What is the precise definition of "Disparate impact" used in the experiments section? - Your definition of “bias” in the sense of group-specific thresholds isn’t consistent with the discussion of fair classification/optimal threshold rules. i.e., the optimal fair decision rule under those fairness constraints would have the judges applying group-specific thresholds. So in what sense is setting unequal thresholds in your simulation “biased”? - How does the performance degrade with the fraction of “biased” judges, and the degree of bias? You compare in the COMPAS example for one “bias” level and 0 vs 50% “biased” judges, but I’m not sure if this is presenting a complete picture. In what situations will the proposed strategy lead to improvement? - When presenting the bar charts for the 50% biased case, how are infeasible solutions being treated? Are you reporting the DI/utility omitting any infeasible solutions? How big are the error bars on the utilities? - By what mechanism is the DI being reduced in the experiments? Is it because release rates are changing considerably across the Random, Optimal, Known and Unknown scenarios? I’m especially interested in the answer to this question for the Random vs. Unknown scenario case. If “release” rates are changing significantly, I don’t feel that the empirical evaluation is presenting an apples-to-apples comparisons.

Reviewer 2



This is an interesting paper that focuses on human decision makers that utilize optimal threshold decision rules, but are biased against protected groups (somewhat similar to https://doi.org/10.1109/JPROC.2016.2608741). The focus of the paper is to develop bipartite matching algorithms to help reduce the overall bias of the decision making in a sequential decision making setup. The paper is interesting and quite a worthwhile contribution that should be highlighted. This is a new perspective on fairness and bias that has not really been discussed before --- although it seems to me that it is a much broader contribution than the specifics of fairness that the authors have couched it in. The definitions and math all appear correct to me. The experiments are sufficient to validate the idea, and look to be conducted properly. My main hesitation regarding the paper is that it may not fully connect to all relevant literature. Matching algorithms are considered in crowdsourcing task allocation problems (here is one example paper: http://wiser.arizona.edu/papers/YANG-CNF-SECON2017.pdf), and that literature should be discussed. The general idea of sequential detection and more general decision making, which goes back to the work of Wald in the 1940s should be discussed as it is very much related to this paper. More broadly, the work should be contextualized within the literature of statistical signal processing. Line 188: "where and m_z(t)" <--- the "and" seems to be a typo

Reviewer 3



Summary/Contribution This paper proposes methods for matching human decision makers to decision subjects, in order to optimize the decisions made (optionally, under fairness constraints). Assuming judges apply threshold rules (where the thresholds differ by judge and—potentially—race), the authors develop efficient matching algorithms even when the thresholds are initially unknown. This paper introduces an interesting research direction to the fairness community. Using algorithms to improve human decision making is an important and under-studied problem, since human decision makers won’t be replaced by algorithms any time soon. However, unsatisfying experiments and a lack of ethical grounding stop this paper from being a clear accept. Weaknesses: In a number of experiments, the “known theta” condition has a higher utility than the “optimal” condition. As I understand it, the “optimal” rule assumes judges all apply the optimal threshold rule, while under “known theta” the judges apply a variety of suboptimal thresholds. How can the latter ever outperform the former? I would like to see the authors examine the ethical implications of their algorithms. I presume that satisfying fairness constraints means assigning all the harsh judges to defendants from one race, while assigning all the lenient judges to defendants from the other race. This is pretty clearly discriminatory, and nicely demonstrates the problem with trying to satisfy common notions of fairness. The authors claim that “higher diversity across the pool experts helps,” but what this means is “it’s good to have some really strict judges to assign to race A, and some really lenient judges to assign to race B,” which doesn’t actually seem desirable. Finally, I didn’t find the results in Figure 3 compelling. With 3x more judges than decisions in a round (and only 50% of judges biased), we can ensure than every decision is made by an unbiased judge (although if theta is unknown it may take us some time to determine who is unbiased). In practice, these decisions are constrained by the number of judges, so N = m and all judges will have to be used. Clarity: The paper is generally well written and clear, with only a typo or two. I found their results to be clearly explained, even though I'm not an expert in graph theory or reinforcement learning (so I couldn't check their math). Originality: This paper is very original. To my knowledge no other paper has proposed matching decision makers to individuals to improve decision making. Significance: Since many decisions will continue to be made by humans, the unconstrained algorithm proposed in this paper has the potential to see real-world implementation. However, no real-world decision maker would assign harsh/lenient judges by race, so a large part of the paper has minimal significance. Response to author rebuttal: I'm glad to see the authors will investigate the ethics of their approach in more detail in the revised paper. I am concerned, though, that the proposed modification (constraining different groups to be assigned to harsh and lenient judges at similar rates) will simply result one group being assigned to the harshest judges within the "harsh" and "lenient" subsets. Once you commit to removing disparate impact as defined in this paper, you commit to holding some groups to lower standards. This is the fundamental ethical problem with trying to remove DI, and I don't see how the method is this paper could provide a way out. As a way to improve the efficiency of human decisions and mitigate the effects of biased human decision makers, I believe this paper has a lot of merit. I would focus the paper on these goals, and leave out the discussion of disparate impact.